# MoWM: Mixture-of-World-Models for Embodied Planning via Latent-to-Pixel Feature Modulation

## Abstract

Embodied action planning is a core challenge in robotics, requiring models to generate precise actions from visual observations and language instructions. While video generation world models are promising, their reliance on pixel-level reconstruction often introduces visual redundancies that hinder action decoding and generalization. Latent world models offer a compact, motion-aware representation, but overlook the fine-grained details critical for precise manipulation. To overcome these limitations, we propose MoWM, a mixture-of-world-model framework that fuses representations from hybrid world models for embodied action planning. Our approach uses motion-aware representations from a latent model as a high-level prior, which guides the extraction of fine-grained visual features from the pixel-space model. This design allows MoWM to highlight the informative visual details needed for action decoding. Extensive evaluations on the CALVIN benchmark demonstrate that our method achieves state-of-the-art task success rates and superior generalization. We also provide a comprehensive analysis of the strengths of each feature space, offering valuable insights for future research in embodied planning.

## 1 Introduction

Embodied action planning represents a core research direction in embodied intelligence, aiming to enable robots to generate precise executable actions from environmental observations and language instructions (Ma et al., 2024; Fung et al., 2025). Early methods primarily relied on imitation learning (IL) (Jang et al., 2022; Chi et al., 2023) from expert demonstration trajectories; however, such approaches often exhibit limited generalization and struggle to adapt to novel scenarios. With recent advances in large models, Vision-Language-Action (VLA) models (Kim et al., 2024; Black et al., 2024; Cheang et al., 2025) have emerged as a promising alternative, offering enhanced capabilities in complex task understanding. Despite these improvements, their training paradigm remains fundamentally based on imitation learning and relies on high-quality demonstration data and faces challenges in achieving broad generalization. In parallel, another line of research explores video-based world models (Hu et al., 2024; Feng et al., 2025; Liao et al., 2025) for action planning. This paradigm involves pre-training a world model on large-scale video datasets to learn general physical dynamics, followed by establishing a mapping between visual observations and robot actions. This approach offers greater data efficiency and is expected to have potential for cross-domain generalization acquired from rich video dynamics learning.

Despite its promise, a key limitation of current world model-based embodied planning lies in the visual representation learning. These models typically rely on features from diffusion-based video generation models (Blattmann et al., 2023; Yang et al., 2024; Wan et al., 2025) pre-trained with pixel-level reconstruction objectives. Such objectives emphasize detailed pixel recovery, yet many robotic tasks do not require perfect reconstruction of all visual elements (Assran et al., 2025). Uniformly encoding all pixels may introduce irrelevant signals that hinder action decoding, complicate the learning of vision-action mappings, and hurt generalization due to overfitting to some task-irrelevant visual details. Alternatively, another line of research explores latent world models (Assran et al., 2025; Zhou et al., 2024; Baldassarre et al., 2025), which learn state transitions in a compressed feature space rather than reconstructing raw pixels. These models employ an encoder to project

video sequences into a compact latent representation and a predictor to model dynamics in this space. By focusing on learning motion-aware representations, such methods are more suitable for guiding global action planning. However, they may overlook fine-grained visual details, potentially leading to inaccuracies in tasks requiring dense object interaction.

To address the redundancy of visual representations in video world models, we propose MoWM, a hybrid world model framework for effective embodied planning. Our motivation is to leverage the latent world modeling features as a high-level prior to guide the extraction of fine-grained visual knowledge from the pixel space. This not only eliminates redundant low-level visual information but also avoids the loss of crucial details when using coarse-grained latent features alone. To be specific, our method consists of two stages. In the first stage, we individually train a pixel world model based on video diffusion and a latent world model on embodied manipulation data. Both models are trained to predict future states in their respective spaces conditioned on text instructions. In the second stage, we combine the two world models. The latent world model's representations are used to modulate the pixel world model's representations, yielding a fused motion-aware low-level visual representation. In this way, we can direct attention to the visual information most relevant to action decoding, which is then fed into an inverse dynamics model for end-to-end action decoding.

We evaluate our approach on the standard embodied manipulation benchmarks CALVIN (Mees et al., 2022), comparing it against imitation learning-based, VLA-based and world model-based action planning methods. Experimental results demonstrate that our proposed MoWM achieves state-of-the-art performance in task success rates, highlighting the significant potential of hybrid world modeling for embodied action planning. Furthermore, we provide an in-depth analysis of the pixel-level and latent-level visual features during action planning, offering practical guidance for model selection in real-world applications.

The main contributions of this work are summarized as follows:

- We propose MoWM, a hybrid world model architecture for embodied action planning, which integrates the motion-aware advantages of a latent world model with the low-level detail generation capabilities of a pixel space world model.
- We explored the interactions of visual features from both pixel and latent space world models. We propose an innovative fusion scheme and offer a comprehensive analysis of its effectiveness.
- Extensive experimental results demonstrate the superiority of our proposed method in both task success rate and generalization to unseen scenarios in embodied action planning.

## 2 RELATED WORKS

### 2.1 VISION-LANGUAGE-ACTION MODELS FOR EMBODIED PLANNING

Vision-Language-Action (VLA) models, which use a large language model backbone enhanced with a vision encoder and an action decoder to predict executable robot actions based on the text instruction, current observation and robot state. Representatively, Pi0 (Black et al., 2024) uses a pre-trained VLM as its foundation and adds an action expert to map VLM tokens to the action space, which is trained with a flow matching objective. Octo (Team et al., 2024) employs a transformer-based LLM to deal with interleaved language, visual observation and action tokens, enabling it to flexibly adapt to new observations and action types. More recent works have focused on improving the complex planning and reasoning ability of VLAs (Zhao et al., 2025; Intelligence et al., 2025; Huang et al., 2025). For example, CoT-VLA (Zhao et al., 2025) introduces intermediate thinking steps such as goal state prediction to enhance action planning. Despite these advancements, VLA models face several limitations. The high cost of collecting teleoperation data makes it difficult to cover a wide range of tasks and diverse scenarios, leading to poor generalization beyond the training environment. Furthermore, their reliance on imitation learning limits their ability to perform counterfactual reasoning or handle complex tasks.

### 2.2 WORLD MODELS FOR EMBODIED ACTION PLANNING

To mitigate the reliance of imitation learning on high-quality interaction data, recent research has explored the paradigm of embodied action planning based on world models (Hu et al., 2024; Liao

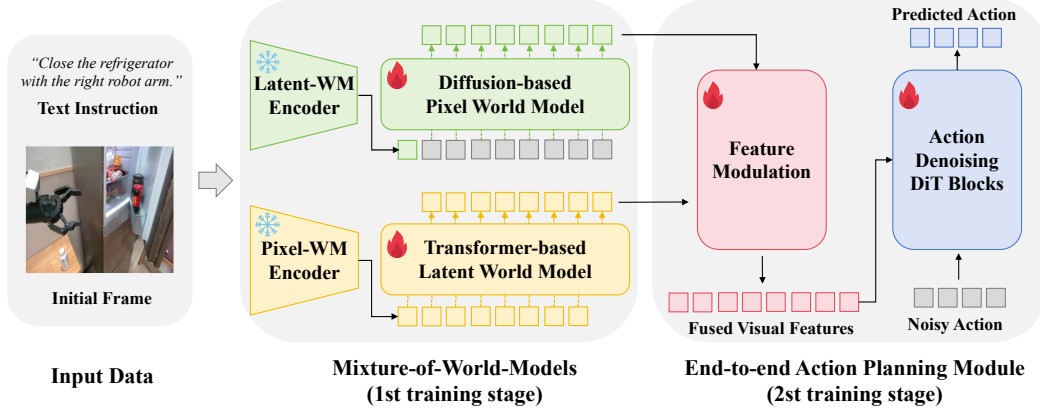

Figure 1: Overall framework of MoWM. In the first stage, we independently train a pixel-space and a latent-space world model driven by text and an initial frame. In the second stage, we freeze the world models, perform latent-to-pixel feature modulation, and then end-to-end train an action denoising network for action planning.

et al., 2025; Feng et al., 2025; Chi et al., 2025; Shang et al., 2025). World models are trained in an unsupervised manner on large-scale video data to learn universal dynamics for downstream tasks (Ding et al., 2024). In embodied action planning, two primary approaches have emerged: the first maps predicted future state sequences to action sequences via an inverse dynamics model in an end-to-end manner (Hu et al., 2024; Feng et al., 2025), which often requires additional adaptation and fine-tuning for specific robot embodiments. The second approach leverages a pre-trained action-conditioned world model to sample multiple action trajectories, evaluate the resulting states, and select the trajectory that maximizes a reward function (Assran et al., 2025; Bar et al., 2025). While more straightforward, this sampling-based method suffers from computational inefficiency and inferior accuracy compared to end-to-end learning. Accordingly, our work adopts the former paradigm and introduces a dedicated action decoder to infer actions from future state predictions generated by the world model.

## 3 METHODOLOGY

World models pre-trained on large-scale video data exhibit remarkable capabilities in predicting future dynamics, where the forecasted states inherently encapsulate rich action-oriented information. This enables such models to serve as powerful priors for guiding action planning. Current approaches in this paradigm primarily leverage video diffusion models as world models (Feng et al., 2025; Hu et al., 2024), extracting intermediate features that capture fine-grained low-level visual details. However, these features often contain substantial noise and irrelevant information (e.g., static background elements), which may hinder action decoding. To address this, we propose modulating the low-level features from pixel-based world models with a latent-space world model specifically designed to learn global temporal dynamics. This hybrid integration enhances action-relevant signals while preserving necessary visual details. The whole framework is illustrated in Figure 1. In this section, we first introduce the pre-training of hybrid world models in Section 3.1, followed by their integration into an end-to-end embodied action planning framework in Section 3.2.

### 3.1 INSTRUCTION-CONDITIONED TRAINING OF HYBRID WORLD MODELS

Effective embodied action planning requires a world model capable of predicting future states from an initial observation based on a natural language instruction. We formalize this core capability as text instruction-conditioned future state prediction and approach it by pre-training two complementary world models: one operating in pixel space and the other in a compressed latent space, both incorporating textual conditioning.

For the pixel-space world model, we leverage Stable Video Diffusion (SVD) (Blattmann et al., 2023) as the base model, following the practice in VPP (Hu et al., 2024). This model is initially pre-trained on generic video data and subsequently fine-tuned on embodied domain datasets to enable embodied instruction-following capabilities. We further inject text conditioning via cross-attention to guide the generation process. The model's primary task is to generate a future video sequence, denoted as $\{x_t\}_{t=1}^T \in \mathbb{R}^{T \times H \times W \times 3}$, given an initial frame $x_0 \in \mathbb{R}^{H \times W \times 3}$ and a language instruction $l$. This is achieved by iteratively denoising a sequence of noisy latents, $\{z_t\}_{t=1}^T \in \mathbb{R}^{T \times C \times h \times w}$, while conditioning on the encoded initial frame and text. The core diffusion process is to learn a denoising function $\epsilon_\theta$ that predicts the noise $\epsilon \in \mathbb{R}^{C \times h \times w}$ added to the latent representation at each time step. The objective is to minimize the following loss:

$$\mathcal{L}_{\text{Pixel-WM}} = \mathbb{E}_{t,x_0,l,\epsilon}[||\epsilon - \epsilon_\theta(z_t, x_0, l, t)||_2^2], \tag{1}$$

where $z_t \in \mathbb{R}^{C \times h \times w}$ is the noisy latent representation of a future frame $x_t$, and the denoiser $\epsilon_\theta$ is parameterized by $\theta$. Here, $H, W$ are the height and width of the video frames, while $h, w, C$ are the height, width, and channel count of the latent space representations.

For the latent-space world model, we first leverage a pre-trained encoder $\boldsymbol{E}(\cdot)$ implemented with the ViT-g from V-JEPA 2 (Assran et al., 2025), which tokenizes each video frame $x_i \in \mathbb{R}^{H \times W \times 3}$ into a sequence of visual tokens $s_i \in \mathbb{R}^{N_s \times D}$. We then introduce a transformer-based latent world model $\boldsymbol{F}(\cdot)$ designed to forecast future states within this latent space. The model is composed of a series of transformer blocks, where each block includes an attention layer and a SwiGLU-activated feed-forward network (FFN). The input to the model comprises the encoded text instruction tokens $c \in \mathbb{R}^{N_c \times D}$ and state tokens from current and past frames $\{s_j\}_{j \leq k}$, which are concatenated and processed to predict the next state token $\hat{s}_{k+1} \in \mathbb{R}^{N_s \times D}$. The model is trained using a teacher-forcing strategy with an L1 loss:

$$\mathcal{L}_{\text{Latent-WM}} = \mathbb{E}_{k,c,s}\left[||\boldsymbol{F}(c, \{s_j\}_{j \leq k}) - s_{k+1}||_1\right]. \tag{2}$$

This model is trained on the same embodied datasets, enabling it to anticipate world state transitions from language instructions directly in the latent space.

## 3.2 END-TO-END ACTION PLANNING VIA MIXTURE-OF-WORLD MODELS

Following the training of our pixel-space world model $G(\cdot)$ and latent-space world model $\boldsymbol{F}(\cdot)$, we exploit their predictive capabilities to guide action generation. A key insight is to use the motion awareness captured by the latent world model's representations to modulate and enhance the features extracted by the pixel-space world model.

Our framework takes an initial frame $x_0 \in \mathbb{R}^{H \times W \times 3}$ and a language instruction $l$ as input. Both world models perform a single forward pass to generate a sequence of features for $T$ future time steps. For the pixel-space world model, we adopt a single-step denoising process inspired by VPP (Hu et al., 2024) to efficiently generate a rich, multi-scale visual representation. Specifically, we extract feature tensors $\{\boldsymbol{V}_i\}_{i=1}^n$ from $n$ distinct upsampling layers of the U-Net. Each tensor $\boldsymbol{V}_i \in \mathbb{R}^{T \times h_i \times w_i \times C_i}$ is a different scale. We then apply a bilinear upsampling operation $\mathcal{U}_{h,w}(\cdot)$ to each tensor to unify their spatial dimensions to $(h, w)$. Finally, these upsampled features are concatenated channel-wise to form a single, aggregated low-level visual feature tensor $\boldsymbol{\Phi}_{\text{pixel}} \in \mathbb{R}^{T \times N_s \times C_{\text{low}}}$:

$$\boldsymbol{\Phi}_{\text{pixel}} = \text{Concat}\left(\mathcal{U}_{h,w}(\boldsymbol{V}_1), \ldots, \mathcal{U}_{h,w}(\boldsymbol{V}_n)\right), \tag{3}$$

where $C_{\text{low}} = \sum_{i=1}^n C_i$ and $N_s = h \times w$. This allows us to capture diverse visual information at different resolutions within a single feature tensor, which is then used as the low-level feature for subsequent action decoding. For the latent-space world model, its state transitions are inherently modeled in the latent space. Thus, we directly extract its output feature sequence $\boldsymbol{\Phi}_{\text{latent}} \in \mathbb{R}^{T \times N_s \times C_{\text{latent}}}$. To align both feature streams for fusion, we apply a linear projection to map them to a shared embedding dimension $D$. The aligned feature tensors, $\boldsymbol{\Phi}'_{\text{pixel}} \in \mathbb{R}^{T \times N_s \times D}$ and $\boldsymbol{\Phi}'_{\text{latent}} \in \mathbb{R}^{T \times N_s \times D}$, are denoted as:

$$\boldsymbol{\Phi}'_{\text{pixel}} = \mathcal{W}_{\text{pixel}}\boldsymbol{\Phi}_{\text{pixel}}, \quad \boldsymbol{\Phi}'_{\text{latent}} = \mathcal{W}_{\text{latent}}\boldsymbol{\Phi}_{\text{latent}}, \tag{4}$$

where $\mathcal{W}_{\text{pixel}}$ and $\mathcal{W}_{\text{latent}}$ are distinct learned projections.

After obtaining hybrid features from the two world models, we perform feature fusion by first concatenating the motion-aware latent features $\Phi'_{\text{latent}}$ and the low-level pixel features $\Phi'_{\text{pixel}}$. This concatenated representation is then passed through a linear projection layer, yielding the fused feature representation $\Phi_{\text{fused}} \in \mathbb{R}^{T \times N_s \times D}$, which is subsequently used by the action planning module. The fusion process is defined as:

$$\Phi_{\text{fused}} = \text{LinearProjection}\left(\text{Concat}(\Phi'_{\text{latent}}, \Phi'_{\text{pixel}})\right). \tag{5}$$

The final aggregated feature for action decoding is generated through a learnable residual mechanism, which integrates the fused feature with the low-level pixel feature. This design enables the model to preserve fine-grained visual details while benefiting from the high-level semantic context captured by the fused representation. The final output visual feature is expressed as:

$$\Phi_{\text{final}} = \mathcal{W}_{\text{gate}}\Phi_{\text{fused}} + \Phi'_{\text{pixel}}, \tag{6}$$

where $\mathcal{W}_{\text{gate}}$ is a learnable gating matrix.

After obtaining the visual feature of future states, we adopt a Diffusion Policy (Chi et al., 2023) as the action decoding module. The fused feature $\Phi_{\text{fused}}$ serves as the condition to guide the multi-step denoising of an initially noisy action vector. The denoiser $\epsilon_\theta$ progressively refines a noisy action vector $a_t \sim \mathcal{N}(0, I)$ based on the fused features to produce the final predicted action. The denoising loss function is denoted as follows:

$$\mathcal{L}_{\text{denoise}}(\psi) = \mathbb{E}_{a_0, \epsilon, k}\left[||\epsilon - \epsilon_\theta(a_t, \Phi_{\text{final}}, t)||_2^2\right]. \tag{7}$$

## 4 EXPERIMENTS

### 4.1 EXPERIMENTAL SETUP

**Datasets.** We evaluate our model and baselines on the CALVIN dataset (Mees et al., 2022), a benchmark designed for long-horizon, language-conditioned robot manipulation tasks. The evaluation focuses on onboard operation with a 7-DOF Franka Emika Panda robot arm. Following established settings (Wu et al., 2023; Hu et al., 2024), we exclusively train our model on data with language instructions and evaluate its generalization ability in the ABC→D split. In this setup, the model is trained on a combination of three scenes (A, B, and C) and tested on an unseen scene (D). This specific task configuration effectively assesses the model's capacity for robust action planning and its ability to generalize to novel environments.

**Baselines.** We compare our approach against three representative categories of embodied action planning models: imitation learning-based methods, VLA-based methods and world model-based methods. All models are fine-tuned on the used dataset for fair comparisons. Details of baselines are introduced as follows:

- **RT-1** (Brohan et al., 2022): This method utilizes a Transformer to map observation images and language instructions to discrete robot actions, trained on expert trajectories based on imitation learning.

- **Diffusion Policy** (Chi et al., 2023): This method learns to denoise action vectors using a diffusion model, with visual observations and poses injected via cross-attention for end-to-end action prediction.

- **3D Diffusor Actor** (Ke et al., 2024): This model integrates 3D scene perception and language instructions for action diffusion denoising.

- **RoboFlamingo** (Li et al., 2023): This method pre-trains a VLM for visual-language understanding and then fine-tunes it with an action head for action prediction.

- **3D-VLA** (Zhen et al., 2024): This model employs a series of 3D perception and generation auxiliary tasks during training to enhance VLA's perception, reasoning, and generation performance in embodied scenarios.

- **OpenVLA** (Kim et al., 2024): This model integrates DINO and SigLIP visual features into a pre-trained LLM, trained on a large-scale dataset of real-world robot manipulation trajectories.

Table 1: Comparison of various embodied action planning methods on the CALVIN dataset. We report the $i$-th task success rate and the average length (in steps) of successful tasks.

| Category | Method | $i^{th}$ Task Success Rate | | | | | |
|---|---|---|---|---|---|---|---|
| | | 1 | 2 | 3 | 4 | 5 | Avg. Len ↑ |
| Imitation learning-based | RT-1 | 0.533 | 0.222 | 0.094 | 0.038 | 0.013 | 0.90 |
| | Diffusion Policy | 0.402 | 0.123 | 0.026 | 0.008 | 0.000 | 0.56 |
| | 3D Diffusor Actor | 0.922 | 0.787 | 0.639 | 0.512 | 0.412 | 3.27 |
| VLA-based | Robo-Flamingo | 0.824 | 0.619 | 0.466 | 0.331 | 0.235 | 2.47 |
| | 3D-VLA | 0.447 | 0.163 | 0.081 | 0.016 | 0.000 | 0.71 |
| | OpenVLA | 0.913 | 0.778 | 0.620 | 0.521 | 0.435 | 3.27 |
| | Pi0 | 0.938 | 0.850 | 0.767 | 0.681 | 0.599 | 3.92 |
| World model-based | Susie | 0.870 | 0.690 | 0.490 | 0.380 | 0.260 | 2.69 |
| | Uni-Pi | 0.560 | 0.160 | 0.080 | 0.080 | 0.040 | 0.92 |
| | GR-1 | 0.854 | 0.712 | 0.596 | 0.497 | 0.401 | 3.06 |
| | Vidman | 0.915 | 0.764 | 0.682 | 0.592 | 0.467 | 3.42 |
| | VPP | 0.909 | 0.815 | 0.713 | 0.620 | 0.518 | 3.58 |
| | **MoWM** | **0.943** | **0.873** | **0.812** | **0.750** | **0.675** | **4.10** |

- **Pi0** (Black et al., 2024): This model uses a pretrained VLM and adds an action expert trained with a flow matching objective.

- **Susie** (Black et al., 2023): This model first generates a goal image using an image editing model and then trains a goal-conditioned policy for action planning.

- **Uni-Pi** (Du et al., 2023): This method first uses a text-driven diffusion model for video prediction, followed by an inverse dynamics model for action decoding.

- **GR-1** (Wu et al., 2023): This model is trained under a multi-task learning paradigm on a large-scale dataset of embodied manipulation videos, enabling it to simultaneously generate future images and plan actions in an end-to-end manner.

- **Vidman** (Wen et al., 2024): This is a two-stage method that begins with video generation pre-training on embodied video data and then adapts the pre-trained model for action planning by adding a self-attention adapter.

- **VPP** (Hu et al., 2024): This is an action planning method based on a video generation diffusion model. It is first pre-trained for text-to-video generation, and its intermediate visual features are then connected to an action decoding module for end-to-end action planning. We use the single-view version as our implementation.

**Implementations of MoWM.** Our framework integrates two distinct world models. The latent world model is built upon the ViT-g encoder from V-JEPA 2 (Assran et al., 2025) and comprises a 24-layer transformer network with approximately 400M parameters. We trained the latent world model on a distributed setup utilizing four H20 GPUs. The training process was completed in approximately 7 hours, running for 75 epochs. Each epoch consisted of 300 steps. For our distributed training setup, we configured a batch size of 4 per GPU, resulting in an effective global batch size of 16. Data loading was parallelized across 12 workers to ensure efficient throughput. For optimization, we used the AdamW optimizer with a cosine learning rate scheduler. A weight decay of 0.04 was applied, which was also annealed to a final value of 0.04 over the full training duration. The pixel world model is the SVD model as implemented in VPP (Hu et al., 2024). For the second stage, we trained our end-to-end action planning module on four NVIDIA H20 GPUs for 13 hours. This stage involved training for 7,000 steps with a batch size of 28. The optimizer for this stage was AdamW, using a learning rate of 1e-4 with a weight decay of 0.05.

## 4.2 MAIN RESULTS

**Quantitative result comparison**. We present a quantitative comparison between MoWM and several baseline methods in Table 1, which reports task success rates on the CALVIN benchmark. Each evaluation task consists of a sequence of five sub-tasks. We report the success rate for each stage

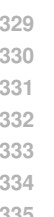

**"Take the blue block and rotate it to the right"**

**"Go push the pink block right"**

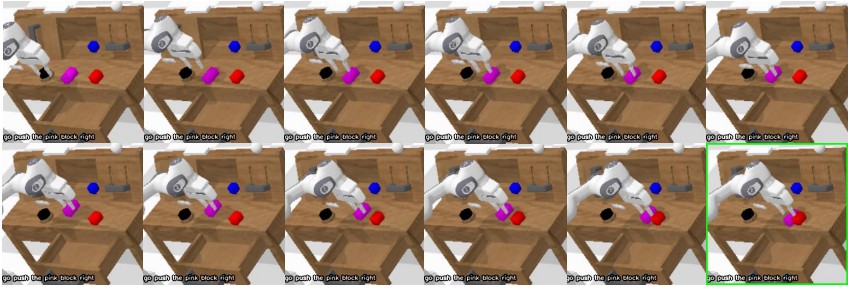

Figure 2: Illustration of the execution process of our model's planned actions in the simulation environment of CALVIN.

and the average completed task length. Across all metrics, our model achieves state-of-the-art performance, validating the effectiveness of the proposed mixture of world models for action planning. MoWM achieves 5.7%, 13.4% improvement in the averaged task success rate across all five stages compared to the most competitive VLA-based and world model-based baselines. Furthermore, by comparing different types of methods, we observe that both VLA-based and world model-based approaches generally outperform imitation learning. This suggests that incorporating complex reasoning or future state prediction is beneficial for improving success rates in embodied manipulation. The performance of VLA and world model methods is broadly comparable. Specifically, while existing world model methods rely on explicit future state generation through image editing or video diffusion, their performance can be compromised by the quality of generated images. Our approach, by contrast, enhances visual feature learning by introducing a latent-space world model, which reduces the reliance on pixel-level features and contributes to more accurate action planning.

Notably, our method exhibits a **stronger performance advantage in long-horizon tasks**, achieving a 12.7% improvement on the 5th task success rate compared with the most competitive baseline. This indicates that the mixture-of-world-models framework excels at capturing extended action patterns. By incorporating future state reasoning, our method mitigates the tendency of imitation learning and VLA approaches to become trapped in local optima due to their over-reliance on immediate observations, thereby facilitating more effective long-range action planning. Furthermore, we provide qualitative evidence of our model's performance. Figure 2 presents the execution of our model's planned actions in the simulation environment, demonstrating its ability to generate plausible and effective actions across a variety of scenarios.

**Qualitative results of future state prediction of world models**. To provide an intuitive understanding of our world models' predictive power, we've visualized the future states predicted by both our latent-space and pixel-space models. For our pixel-space world model, we directly show the future video frames generated through its diffusion denoising process. The latent-space model, however, operates in a non-visualizable latent space. To address this, we trained a dedicated decoder to convert the latent states into RGB images for analysis. This decoder is a convolutional neural network with approximately 8.04M parameters. Its architecture consists of four upsampling modules, each

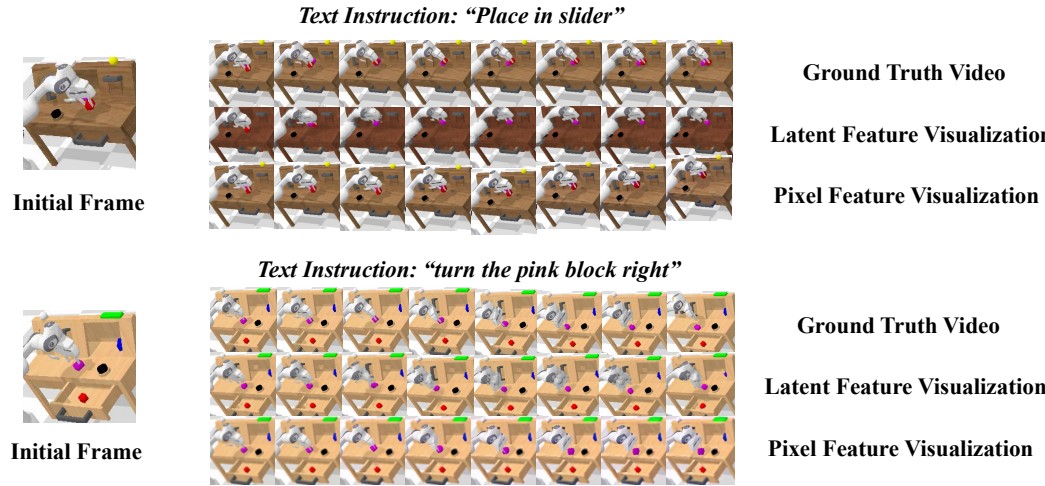

Figure 3: Visualizations of the future state predictions of the latent world model, pixel world model, and the ground truth video.

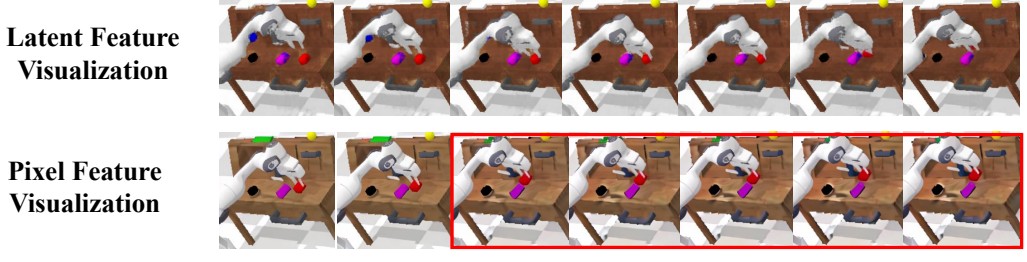

Figure 4: Visual feature comparisons during action prediction rollouts. The pixel world model sometimes produces long periods of static frames, lacking dynamic movement. In contrast, the latent world model consistently exhibits better dynamics, demonstrating its strength in learning and predicting motion.

using a transposed convolution, batch normalization, and a ReLU activation function, followed by a final convolutional output layer. We trained the decoder for 15,000 steps on a single NVIDIA H20 GPU, using 3,200 images ($200 \times 200 \times 3$) from the CALVIN dataset. The training, which took about 30 minutes, used the latent states from our world model's encoder as input and the original images as the target, optimizing with a mean squared error (MSE) loss. Figure 3 showcases the future state predictions from both models. The results indicate that both the latent-space and pixel-space world models can generate plausible future states based on text instructions, containing all the crucial cues needed for action decoding.

## 4.3 ABLATION STUDY

A key design of our framework is to enhance the feature extraction of the pixel-space world model by incorporating representations from a latent-space world model. To evaluate the effectiveness of this design, we conduct ablation studies comparing three model configurations: (1) MoWM (concat-based fusion): a fusion approach where features from both world models are concatenated and then projected; (2) MoWM (cross attention-based fusion): a fusion approach integrates features from the latent-space world model via cross-attention; (3) MoWM (without fusion): a baseline version that relies solely on the low-level features from the pixel-space world model.

Table 2: Ablation study of the latent and pixel feature fusion approaches in MoWM.

| Method | $i^{th}$ Task Success Rate | | | | | |
|---|---|---|---|---|---|---|
| | 1 | 2 | 3 | 4 | 5 | Avg. Len ↑ |
| MoWM (concat-based fusion) | **0.943** | **0.873** | **0.812** | **0.750** | **0.675** | **4.10** |
| MoWM (cross attention-based fusion) | 0.936 | 0.836 | 0.748 | 0.665 | 0.573 | 3.80 |
| MoWM (without fusion) | 0.927 | 0.831 | 0.741 | 0.652 | 0.560 | 3.70 |

As summarized in Table 2, our ablation study confirms the effectiveness of fusing latent and pixel-space features. The concat-based fusion model achieves the best performance, followed by the modulation-based variant, with the no-fusion baseline performing the worst. These results validate the clear benefit of leveraging latent-space world model features for representation enhancement. The superior performance of the simple concat operation is particularly notable. By preserving the full relevant visual details from low-level features and the motion-aware information from latent features, it creates a more comprehensive visual representation that is better suited for action decoding. Specifically, the concat-based model outperforms the no-fusion variant by an average of 11.2% in task success rate across all five stages, demonstrating the significant value of our hybrid approach. While the attention-based modulation fusion is also beneficial, its performance is lower than simple concatenation. This suggests that while attention can be powerful, it may be challenging to learn the optimal alignment between the two distinct feature spaces, potentially limiting its effectiveness and efficiency compared to a more direct fusion. The no-fusion version performs poorly because the low-level features from the diffusion model are not inherently aligned with action decoding objectives and often contain substantial noise and irrelevant details.

To intuitively demonstrate how the latent world model enhances the pixel-space world model, we conducted a qualitative analysis. As shown in Figure 4, we observed that during action planning, the pixel-space model sometimes produced long periods of static future-state predictions, which is problematic for action decoding. These issues stem from the diffusion model's training objective, which focuses on fitting all pixels equally, including static background elements, without explicitly emphasizing dynamic motion. This can result in visually plausible but inaccurate motions that negatively impact subsequent action predictions. In contrast, the latent world model learns in a compact space, allowing it to focus more effectively on motion patterns. As illustrated in the examples, this model consistently generates high-quality dynamic patterns and more coherent motions. Consequently, its latent features can guide the low-level pixel features to more accurately reflect the underlying dynamics.

## 5 CONCLUSION AND FUTURE WORK

In this paper, we introduced MoWM, a mixture-of-world model framework for embodied action planning that effectively bridges the complementary strengths of pixel-space and latent-space world models. MoWM addresses the key limitation of visual redundancy in pixel-based models by using high-level, motion-aware representations from a latent world model to guide and modulate low-level feature extraction. This design enables the model to suppress task-irrelevant visual details while preserving fine-grained information critical for precise manipulation, resulting in a more insightful visual representation for downstream action decoding. Extensive experiments on the CALVIN benchmark validate the effectiveness of the proposed method.

In the future, we envision several promising directions for further research. A key improvement is to explore dynamic fusion strategies that can adaptively weigh the contributions of latent and pixel features based on task complexity. Furthermore, we will validate our approach on additional benchmarks and real-world robotic platforms to assess its applicability. Finally, we also plan to extend MoWM into a more generalized framework pre-trained on large-scale, unannotated video datasets, enabling zero-shot transfer to a wider range of embodied tasks.

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

# A APPENDIX

## A.1 THE USE OF LARGE LANGUAGE MODELS (LLMS)

We acknowledge the use of a large language model (LLM) to assist with this paper. The LLM was used solely for auxiliary purposes, specifically for searching and retrieving relevant literature. It was not involved in core tasks such as idea generation, code writing, data analysis, or manuscript preparation.

## A.2 REPRODUCIBILITY STATEMENT

To ensure reproducibility, we've made our code available in the following anonymous repository: `https://anonymous.4open.science/r/MoE-WM-4C43`.

"put the handle to open the drawer"

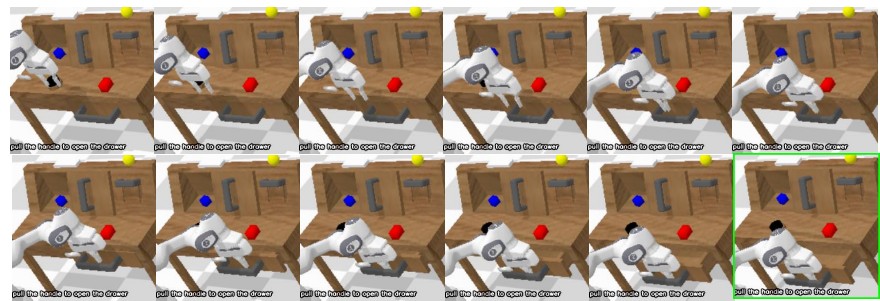

Figure 5: Additional examples of the execution process of our model's planned actions in the simulation environment of CALVIN.

