# OpenReview forum: "MoWM: Mixture-of-World-Models for Embodied Planning via Latent-to-Pixel Feature Modulation"
_ICLR.cc/2026/Conference — Submitted to ICLR 2026_

### Official Review · Reviewer_9HQb · 2025-10-28

**Soundness:** 3
**Presentation:** 3
**Contribution:** 2
**Rating:** 4
**Confidence:** 3

**Summary:**

This paper proposes a mixture-of-world model that integrates pixel-level and latent-level world models. The pixel-level and latent-level features are concatenated and processed in a residual manner (L229). Experiments on the CALVIN benchmark demonstrate that the proposed world model enhances the performance of the diffusion policy, achieving SOTA results on CALVIN.

**Strengths:**

- The idea of combining pixel-level and latent features is compelling.
- The authors propose a simple yet effective strategy to fuse these two distinct types of features.
- Experiments on the CALVIN benchmark demonstrate that the proposed world model helps the diffusion policy achieve SOTA performance.

**Weaknesses:**

- The fusion of pixel-level and latent features appears to be heuristic, and while it proves effective, several questions remain. For instance, why is this particular fusion strategy sufficient for feature integration? Could it be extended to other types of latent representations or real-world images?

- The paper lacks quantitative analysis of the MoWM. The authors only provide quantitative results for the robot policy. It would be valuable to understand the reasons behind the significant performance improvements achieved by using MoWM.

**Questions:**

- What is the potential for applying MoWM in real-world environments? Would it still achieve performance improvements under such conditions?
- In Figure 3 (top) and Figure 4, the decoded latent feature images show desks in different colors, while the bottom part of Figure 3 displays a consistent desk color. What explains this discrepancy?

I would be willing to raise my score if the authors could address these concerns, particularly regarding the relationship between the policy performance improvement and MoWM.

---

### Official Review · Reviewer_azXP · 2025-10-29

**Soundness:** 1
**Presentation:** 1
**Contribution:** 2
**Rating:** 2
**Confidence:** 4

**Summary:**

This paper proposes a new mixture framework by fusing representations from hybrid world models, pixel world model and latent world model. The key idea is to use the motion awareness captured by the latent world model’s representations to modulate and enhance the features extracted by the pixel-space world model.

**Strengths:**

* The approach is simple and straightforward.
* The problem that this paper would like to address is clear. Pixel space has redundant information for embodied tasks.

**Weaknesses:**

* The evaluation environment is too simple. The paper only conducts the experiment on the CALVIN.
* It seems the limitation of pixel world model the paper states, “pre-trained with pixel-level reconstruction objectives. Such objectives emphasize detailed pixel recovery, yet many robotic tasks do not require perfect reconstruction of all visual elements” is not the main problem in this paper? It’s very confusing to state it. In my understanding, the limitation for the pixel world model that the paper would like to address is that the pixel has redundant information that hinders action decoding.
* Not ablation for other feature Modulation methods.

**Questions:**

* What if the motion generated by the latent world model and the video generated by the pixel world model are not aligned, like they generate different trajectories for the robot arm?
* In Figure 2, are the latent-WM encoder and pixel-WM encoder in the wrong place?

---

### Official Review · Reviewer_M6LR · 2025-11-01

**Soundness:** 4
**Presentation:** 3
**Contribution:** 3
**Rating:** 8
**Confidence:** 4

**Summary:**

The paper proposes MoWM (Mixture-of-World-Models), a hybrid framework for embodied action planning that combines a latent-space world model (motion-aware, compact) with a pixel-space diffusion-based world model (fine-grained visual detail). The key idea is to use the latent model’s representations as a high-level prior to modulate features from the pixel model, thereby suppressing visual redundancies while preserving manipulation-critical details. The fused representation is then used in an end-to-end diffusion-based action decoder. Evaluated on the CALVIN benchmark, MoWM achieves state-of-the-art task success rates, especially on long-horizon tasks, and includes ablation studies validating the fusion design.

**Strengths:**

1. Originality: The “mixture-of-world-models” concept creatively bridges latent and pixel paradigms. While hybrid representations exist in other domains, their application to embodied planning via latent-to-pixel modulation is novel.
2. Quality: Rigorous experiments across multiple baselines (IL, VLA, world models), strong ablations, and qualitative visualizations support claims.
3. Clarity: Technical details (e.g., feature alignment, fusion via gating) are explained intuitively. Figures effectively communicate key ideas.
4. Significance: Demonstrates that explicit fusion of complementary world models can outperform monolithic approaches, potentially influencing future architectures in robotics and video modeling.

**Weaknesses:**

1. Limited Real-World Validation: All experiments are in simulation (CALVIN). While common, the absence of real-robot testing leaves open questions about robustness to sensor noise, actuation errors, or domain gaps.

2. Fusion Mechanism Simplicity: The best-performing fusion is simple concatenation, which raises the question: Is the gain primarily from multi-source features, or from the specific modulation design? A more nuanced analysis (e.g., attention maps, feature importance) could strengthen the claim.

3. Computational Cost: Training two large world models (SVD + ViT-g transformer) is resource-intensive. The paper does not discuss inference latency or scalability—important for real-time robotics.

**Questions:**

1. Have you tested MoWM on real robots or sim-to-real transfer settings? If not, what are the main barriers?

2. Why did cross-attention underperform concatenation? Was it due to optimization difficulty, or does it suggest that pixel features already contain sufficient spatial structure?

---

### Meta-Review · Area_Chair_reX2 · 2026-01-10

**Summary:**

The reviewers acknowledge the novelty of the hybrid "mixture-of-world-models" approach and the impressive state-of-the-art results achieved on the CALVIN benchmark. However, the following critical concerns informed the recommendation:

Limited Evaluation Scope: All experiments are conducted solely in simulation on the CALVIN dataset. Reviewers expressed significant doubt regarding the model's robustness to real-world sensor noise, domain gaps, and sim-to-real transfer.

Heuristic Fusion Mechanism: While effective, the fusion strategy is viewed as primarily heuristic. Reviewers noted that simple concatenation outperformed more complex modulation, raising questions about whether the performance gain is due to the specific design or merely the inclusion of multi-source features.

Lack of Component Analysis: The paper lacks sufficient quantitative analysis of the world models themselves, focusing instead on the final policy success rates.

Computational and Practical Barriers: The framework requires training two large-scale world models (SVD and a ViT-g transformer), leading to concerns about high computational costs and inference latency in real-time robotics.

Clarity and Consistency: Concerns were raised regarding visual discrepancies in the decoded latent feature images (e.g., inconsistent desk colors) and potential labeling errors in the architectural diagrams.

**Reviewer Concerns:**

Since the authors did not provide a rebuttal during the official discussion period, all concerns raised by the reviewers remain fully outstanding.

**Reviewer Scores:**

As the authors did not submit a rebuttal or engage in the discussion phase, there was no additional information provided to resolve the reviewers' doubts or clarify the technical concerns raised. Consequently, the scores for all reviewers remain unchanged from their initial assessments.

---

### Decision · Program_Chairs · 2026-01-26

Reject